# Psychological counseling in the Italian academic context: Expected needs, activities, and target population in a large sample of students

**Pasquale Musso**[1], **Gabrielle Coppola**[1], **Ester Pantaleo**[2,3]*, **Nicola Amoroso**[3,4], **Caterina Balenzano**[5], **Roberto Bellotti**[2,3], **Rosalinda Cassibba**[1], **Domenico Diacono**[3], **Alfonso Monaco**[3]

**1** Dipartimento di Scienze della Formazione, Psicologia, Comunicazione, Università degli Studi di Bari Aldo Moro, Bari, Italy, **2** Dipartimento Interateneo di Fisica Michelangelo Merlin, Università degli Studi di Bari Aldo Moro, Bari, Italy, **3** Istituto Nazionale di Fisica Nucleare (INFN), Sezione di Bari, Bari, Italy, **4** Dipartimento di Farmacia-Scienze del Farmaco, Università degli Studi di Bari Aldo Moro, Bari, Italy, **5** Dipartimento di Scienze Politche, Università degli Studi di Bari Aldo Moro, Bari, Italy

☯ These authors contributed equally to this work.

* ester.pantaleo@uniba.it

**Data Availability Statement:** All data files are available on https://figshare.com/s/e24f094a4c4062425060.

## Abstract

University psychological counseling (UPC) is receiving growing attention as a means to promote mental health and academic success among young adults and prevent irregular attendance and dropout. However, thus far, little effort has been directed towards the implementation of services attuned to students' expectations and needs. This work intends to contribute to the existing literature on this topic, by exploring the perceptions of UPC among a population of 39,277 students attending one of the largest universities in the South of Italy. Almost half of the total population correctly identified the UPC target population as university students, and about one third correctly expected personal distress to be the main need that UPC should target. However, a large percentage did not have a clear idea about UPC target needs, activities, and population. When two specific student subsamples were analyzed using a person-centered analysis, namely (i) those who expressed their intention to use the counseling service but had not yet done so and (ii) those who had already used it, the first subsample clustered into two groups, characterized by an "emotional" and a "psychopathological" focus, respectively, while the second subsample clustered into three groups with a "clinical", "socioemotional", and "learning" focus, respectively. This result shows a somewhat more "superficial" and "common" representation of UPC in the first subsample and a more "articulated" and "flexible" vision in the second subsample. Taken together, these findings suggest that UPC services could adopt "student-centered" strategies to both identify and reach wider audiences and specific student subgroups. Recommended strategies include robust communication campaigns to help students develop a differentiated perception of the available and diverse academic services, and the involvement of active students to

**Funding:** The author(s) received no specific funding for this work.

**Competing interests:** The authors have declared that no competing interests exist.

remove the barriers of embarrassment and shame often linked to the stigma of using mental health services.

## 1 Introduction

The most recent statistical data indicate that Italy, among the countries of the European Union (EU), has one of the lowest numbers of graduates [1]. Only 29% of young adults between 25 and 34 have earned a degree in Italy compared to an EU average of 41%. This phenomenon represents an issue from several points of view. At a macrosystemic level, as in most nations with greater industrialization, also in Italy graduates represent the future leaders and decision-makers in the economic, educational, and government fields. Therefore, the higher the number of graduates, the larger can be the basis for selection of positive, constructive, innovative and efficient leaders capable of guaranteeing adequate growth standards for the country. At a microsystemic level, the university path, if experienced actively and successfully, represents for young adults a varied, articulated, and flexible circumstance of confrontation with colleagues and professors. This is a unique occasion to reflect in a more mature and conscious way on one's own future in terms of personal, relationship, and career development. This potential flourishing process, of course, also has a wider social impact; especially young university students at the end of their academic career tend to have an increasingly effective direct influence also on their family members, friends, and community, for example by questioning "biased" opinions, attitudes, and points of view and proposing more rationally and scientifically based alternatives. The cost of a low number of young people having completed tertiary education seems, therefore, to be a significantly negative impact on the societies that are characterized by it. It is important to counteract such a trend.

A low number of graduates might depend on a variety of political, sociocultural and psychoeducational factors. Two main patterns are often observed: young people decide "not" to enroll in any university program (or in any equivalent higher educational program), or they enroll but fail to complete their program and abandon their studies. In Italy, the university access rate for the youngest adults, those under the age of 25, as indicated by the Organization for Economic Co-operation and Development (2017) [2], is 41% against an average of 48% for countries that are part of the organization. In addition, among students enrolled at university one year after graduation, about 6–7% drop out and 8–9% switch universities or degree program (see [3]). For some time now, Italian universities have been trying to equip themselves with centers for guidance and tutoring [4] to reduce both the university access gap and the phenomena of dropout or irregular attendance; these centers offer support and follow students from the last years of secondary school until they enter the labor market. Among the services offered by these centers, psychological counseling represents one of the most important ones, since it is intended as a strategic tool not only to increase the academic success of students already enrolled (reducing university dropout levels) but also to make universities contexts for personal development in order to attract more and more young people engaged in their life and post-diploma career choices (reducing the access gap to university). However, for this perspective to be effective, it is necessary to build university psychological counseling (UPC) services that are able to concretely read students' expectations and meet their needs. This work intends to contribute to the existing literature on this topic by trying to explore how students perceive psychological counseling and to grasp their expectations and needs, using a large population of university students. A particular focus will be directed both to those who express

their intention to use the counseling service but have not yet done so and to those who have already used the service.

## 1.1 The importance of UPC

According to the American Psychological Association [5], psychological counseling is a health practice and service that focuses on how people function both personally and in their relationships. A wide range of problems (e.g., emotional, social, work, school, and physical health problems) are targeted by this practice at different stages of human development, with individual, group, and organizational intervention strategies in different contexts, including the educational one [6].

UPC also falls within this definition, although obviously it is characterized by the management of issues that are particularly relevant to the academic career and the age of the end users, i.e., students. In itself, a student's career is full of challenges, such as adjusting their study method, adapting to a complex context with new rules relating to colleagues and faculty, or managing the academic load [7]. Frequently, freshmen cope with these tasks alone, as they have moved out of their family home, which might be far away from the college site. Exam anxiety and uncertain expectations towards the future are additional conditions that can create difficulties and cause malaise to develop in students, fostering feelings of helplessness and confusion and questioning of their personal, emotional and relational sphere. Also, the majority of university students are late adolescents and young adults generally between the ages of 17 and 29 (in Italy, about 87%) [8].

At this age, in addition to the challenges of higher education, students also experience further challenges related to their development, such as the search for autonomy and the right connection with their family, the search for new friendships and intimate relationships, and the pursuit of personal and professional goals [9]. Epidemiological studies on an international scale have largely established that mental illness generally has its onset in this age group (e.g., [10–12]. Thus, a significant number of late teens and young adults will experience their first psychiatric episode during their university period. In particular, recent meta-analyses have estimated a prevalence of about 34% for anxiety and about 27% for depressive symptoms among college students worldwide [13, 14]. These data have also been confirmed in the Italian context [15]. Using psychological distress as a measure of the mental health of 4,760 Italian university students between the ages of 18 and 35, Porru and colleagues [15] found that approximately 78.5% of respondents had experienced episodes of psychological distress in the last month: mild for 21.3%, moderate for 21.1%, and severe for 36.1%. Thus, more than a third of the students had experienced high levels of psychological distress in the previous month. Additionally, women (39.1%) were more likely to exhibit severe levels of psychological distress than men (26.8%).

Taken together, the aforementioned considerations and findings clearly highlight the importance of the availability of effective counseling and mental health services within universities. In the context of a prevalence of mental health problems among young people and college students. which has increased exponentially over the past decade and has increasingly become a topic of national concern [16], such services may represent a relevant and accessible opportunity to oppose this trend. This is all the truer as previous studies suggest that university counseling has been shown to reduce symptoms of psychological distress by approximately the same magnitude as treatments practiced in randomized controlled trials [4, 17]. Nonetheless, the same reflections and the same research outcomes must lead university systems to ask themselves why, however, such a high number of university students continue to show signs of psychological distress and mental illness and how to improve the quality of services.

## 1.2 Use of UPC and the need to characterize students

As already mentioned, the international literature suggests that generally at least 20–30% of university students report symptoms consistent with a diagnosis of mental health problems, such as, for example, self-harm, anxiety, and mood disorders [18]. In addition, some in-depth studies have pointed out that, of these students, about 65% have never received mental health services for their problems [10, 19]. This appears to be in line with what happens in the general population, where about 70% of people have never received mental health services in the face of surveys indicating significant symptoms of mental illness [20]. However, the university context should represent a better opportunity both for students to contact and use internal counseling services and for these same services to raise awareness among their recipients. Closing the gap between the need for counseling for mental health problems and the effective use of the counseling service is therefore one of the most important challenges of university systems, as successful solutions to such problems could have important implications for the well-being of students, for their academic success, for the quality of the universities they attend [4], and ultimately, as mentioned at the beginning, for the growth of the societies and countries in which they are located. Indeed, beyond the need to address human distress and suffering, there is evidence that the mental health problems of university students are negatively linked to further education [21].

Among the possible strategies to shorten the gap between the number of students who would benefit from using a counseling path and those who actually use it, we typically refer to awareness-raising strategies (e.g., [7]). This approach, however, is well known to the various operators of UPC services as well as to the decision-making centers of the university systems; therefore, considered alone, it does not seem to be a sufficiently explanatory dimension. Scarcely explored, however, is the characterization of students in two different ways: first, in terms of general students' expectations about UPC services; second, and most importantly, in terms of differentiation with respect to these expectations between students who, while demonstrating a positive attitude towards UPC services, eventually use them or do not [18, 22]. Such knowledge would have several advantages: (a) it would facilitate the construction of more targeted student awareness strategies, promoting higher rates of UPC services use; (b) it would favor interventions more targeted to the needs of university students, identifying those individuals who would most likely benefit from a counseling path [18, 23]; and (c) it would promote an effective strategy for evaluating UPC and mental health services as well as the further planning and development of medium- and long-term programs of such services.

## 1.3 Research questions and approach

In light of the above considerations, the general purpose of this study was to characterize students' perception of UPC services. Given the exploratory nature of our research, no hypotheses have been formulated; however, we have identified two research questions.

Research Question 1: In the general student population, what are the perceptions and expectations related to (a) those who will use the UPC service (expected target students), (b) the needs to which the UPC service responds (expected target needs), and (c) the activities proposed by the UPC service (expected target activities)? Research Question 2: What are the characteristic features of students who, despite having expressed a positive intention to use the UPC service, did not use it compared to those who did use it?

To answer these two questions, we integrated an observational study with a person-centered approach. The latter has emerged from a holistic-interactionist perspective [24] and has numerous advantages for understanding the experiences of individuals. Person-centered approaches, in fact, emphasize understanding the individual as a whole rather than focusing

on singular characteristics. Therefore, this approach and associated analyses examine, rather than ignore, nonlinearity and interactions between variables [25].

Consequently, typologies derived from person-oriented approaches can identify the co-occurrence of different aspects that characterize individuals, recognizing the "tendency for a given person to have a distinct pattern of factors on which they are high, medium, or low" [26]. In this line, to answer research question 2, we sought to group students with similar response configurations to questions related to the three expectancy variables (expected target students, expected target needs, and expected target activities). These variables, in fact, can be thought of as "nonlinear" in their relations to each other and to the outcome of actually using or not using the UPC. For example, in one of the authors' practices of managing UPC, expectations about the type of student who accesses the services are not necessarily proportional (and, therefore, linear) to expectations about the type of needs UPC should address or the type of activities that should be provided.

Considering this, the present study has significant ecological validity, characterized by a complex, dynamic and naturalistic-based collection of information. A good part of the other studies on university counseling, on the other hand, usually prefer the search for factors that, through a quantitative approach, explain the intentions behind the use of counseling rather than direct perceptions and expectations as well as the behavior of using such a service [27]. To our knowledge, there are only a few studies with a similar approach in the international context (see, for example, [18])and there are none in the Italian one.

## 2 Materials and methods

### 2.1 Participants

The sample consisted of 39,277 students enrolled at the University of Bari Aldo Moro. The main site of the University is located in the city of Bari, while some courses are located in two other cities (Brindisi and Taranto) in the same Region of Italy (Apulia). The sample represented 89.57% of the total student population (N = 43,848) at the time of collection (census on March 31, 2017) and had a mean age of 24 years (SD = 5, in the range 18–70); 62.5% (N = 24,543) were women. Of the collected sample, 71.9% (N = 28,224) were on-track students, that is students who were on time with their academic career; 25.3% (N = 9,925) were off-track students, that is students who would not complete their studies within the standard time; 2.9% (N = 1,128) were inactive students or students who had not acquired study credits or written exams in the last 12 months. As to course type, 60.4% of the sample (N = 23,708) were enrolled in a bachelor's degree (first-level three-year degree or "triennale"), 21.2% (N = 8,337) a master's degree (second-level two-year degree or "magistrale"), and 18.4% (N = 7,232) a single-level degree of five to six years (e.g., Medicine or Primary Education, "a ciclo unico"). Following the classification of the Italian Ministry [28], which classifies university programs within four macroareas, 59% of the sample (N = 23,191) were enrolled in a sociohumanistic degree (e.g., Linguistics, Philosophy, Psychology, Social Sciences), 9.7% (N = 3,815) in a medical degree, 24.5% (N = 9,632) in a scientific-technological degree (Biology, Physics, Statistics, Mathematics), and 6.7% (N = 2,639) in a health degree (e.g., Rehabilitation, Nursing).

### 2.2 Data collection procedure

A questionnaire was uploaded in the students' lifecycle and career personal accounts managed by the Student Management System in use in Italian Universities (Esse3, by CINECA [29]). At login, students were asked to complete the questionnaire before proceeding to other functions (e.g., register for an exam, access documentation and career data, etc.). The questionnaire ran

for 40 days, matching an exam session of the academic year, and had a high number of participants (N = 39,277). Data collected for the current study are publicly available [30]. No personal or sensitive information was asked, and the data was anonymous, as participants were labeled with a progressive number based on the time of access to the survey. The study was approved by the Ethical Committee at the Department of Education, Psychology and Communication at the University of Bari (Ethics reference code: ET-22–03).

## 2.3 Survey structure

The questionnaire was built ad hoc to investigate what students knew about the current counseling service on campus and what they expected of it; it was structured into three sections with both closed- and open-ended questions.

In the first section we collected respondents' general information (sex, age, course, etc.), summarized in Section 2.1 and listed in Table 1. We will refer to these data as *supplementary variables*.

In the second section, we used closed-ended questions to investigate what students knew about the current service on campus and to whom they thought it was directed. Also, we asked whether they ever felt the need to use it, or if they did actually use it. Results are summarized in Section 3.1. We added open-ended questions to collect the motivations behind the selected choices.

**Table 1. Supplementary variables with their respective categories.** This table collects answers to the first part of the survey.

| Categories | | Total sample (N = 39,277) | Subsample 1 (willing to use the counseling service but did not N = 4,440) | Subsample 2 (accessed the counseling service N = 545) |
|---|---|---|---|---|
| **Sex** | | | | |
| 1 | Women | 24,543 | 3,228 | 347 |
| 2 | Men | 14,734 | 1,212 | 198 |
| **Age bin** * | | | | |
| 1 | 17–24 | 28,440 | 2,922 | 366 |
| 2 | 25–29 | 7,182 | 998 | 112 |
| 3 | 30–29 | 2,696 | 413 | 52 |
| 4 | over 40 | 959 | 107 | 15 |
| **Type of degree** | | | | |
| 1 | Single-level university degree of 5 to 6 years | 7,232 | 1,083 | 104 |
| 2 | First-level three-year university degree | 23,708 | 2,445 | 326 |
| 3 | Second-level two year university degree | 8,337 | 912 | 115 |
| **Student status** | | | | |
| 1 | on-track (will graduate on time) | 28,224 | 2,786 | 372 |
| 2 | off-track | 9,925 | 1,410 | 155 |
| 3 | inactive (haven't acquired CFUs, i.e., study credits, or given exams in the last year) | 1,128 | 244 | 18 |
| **Disciplinary area** | | | | |
| 1 | Medical | 3,815 | 670 | 66 |
| 2 | Scientific/Technological | 9,632 | 918 | 99 |
| 3 | Health | 2,639 | 221 | 37 |
| 4 | Socio-Humanistic | 23,191 | 2,631 | 343 |

* students are categorized according to age groups.

In the third section, we investigated the respondents' expectations concerning psychological counseling in the academic context. With a closed-ended question with multiple choices, we investigated expectations about the target population of such a service ("Expected target students"); then, with two open-ended questions we investigated the expected needs such a service should target ("Expected target needs") and the expected activities it should offer ("Expected activities"), respectively. Answers to these open-ended questions were coded using a bottom-up procedure: categories were identified starting from content analysis of the answers and were not theoretically driven. However, for the open-ended question regarding activities, a total of 382 answers (<1% of the sample) presented more than one category; by convention, only the first category emerging from the answer was used in the data analyses. The list of categories corresponding to these three questions is shown in Table 2, and we will refer to the corresponding variables as *active variables*.

## 2.4 Data analysis procedure

We focused our attention on two student subpopulations: students who intended to use the service but had not so far (Group 1) and students who use the service (Group 2). Our aim was to identify subgroups (or clusters) of students in Group 1 with similar answers to the third part of the questionnaire, i.e., to identify subgroups of students that had similar "Expected activities", "Expected target needs", and "Expected target students" whose full list is reported in Table 2. Using the same rationale, we identified subgroups of students in Group 2 as well.

In order to group students who gave similar answers, we used a hierarchical clustering procedure followed by an optimization step based on K-means clustering. As hierarchical clustering needs continuous input data, we first used Multiple Corresponding Analysis (MCA) to transform our input categorical data (the *active variables*) into a set of continuous variables. In other words our "person-oriented" approach statistically consisted of four main steps: (i) Multiple Correspondence Analysis (MCA) to transform categorical variables into a set of continuous variables that explain most of the inertia; (ii) Hierarchical clustering using the obtained continuous variables and Ward's criterion to group students with similar profiles; (iii) choice of the optimal number $K$ of clusters based on the obtained dendrogram; (iv) K-means clustering to improve upon the initial partition obtained in step iii. The data analysis workflow is summarized in Fig 1 and described in the following sections. Furthermore, to characterize the obtained clusters in terms of over- and underrepresented categories, for each category and each cluster, we tested the null hypothesis that the proportion of the population belonging to a category within a cluster is equal to the proportion of the population belonging to the chosen category in the whole population (we used a normal approximation of the hypergeometric distribution of the counts [31]). For our analysis we used R version 4.0.5 [32] and the FactoMineR package [33].

**2.4.1 Multiple correspondence analysis.** Multiple correspondence analysis (MCA) is a multivariate technique that represents an extension of correspondence analysis (CA) to more than two categorical variables [34]. It can also be regarded as a generalization of Principal Component Analysis (PCA) when the data set consists of qualitative instead of quantitative indicators [35, 36]. MCA is used to study relations among categorical variables, in our case the categorical variables are "Expected activities", "Expected target needs", and "Expected target students", and to obtain maps in which to visualize the distances between categories of the input variables or between observations, in our case students. MCA can thus facilitate the interpretation of data structures [37]. MCA is generally used to analyze results of a survey and to group individuals with similar profiles, i.e., individuals who gave similar answers to the survey. In practice, MCA transforms categorical variables into a low dimensional set of

**Table 2. Active variables with their respective categories.** This table collects answers to the third part of the survey and reports for each variable the number of answers in each category with the corresponding percentages (within parentheses).

| Expected activities | | Count (%) |
|---|---|---|
| 1 | Not sure/No answer | 20,702 (52.7) |
| 2 | Sharing and managing personal distress | 7,918 (20.2) |
| 3 | Individual clinical interviews | 3,793 (9.7) |
| 4 | Didactic tutoring | 1,484 (3.8) |
| 5 | Other | 1,442 (3.7) |
| 6 | Group intervention | 1,054 (2.7) |
| 7 | Training on study skills and learning strategies | 971 (2.5) |
| 8 | Empowerment of personal and context adaptation strategies | 759 (1.9) |
| 9 | Insufficient/No knowledge of the service | 662 (1.7) |
| 10 | Orientation and career services | 418 (1.1) |
| 11 | Psychological diagnostic evaluation | 73 (0.2) |
| Total | | 39,277 (100) |

| Expected target needs | | |
|---|---|---|
| 1 | Not sure/No answer | 21,657 (55.1) |
| 2 | Personal experience of emotional distress | 9,173 (23.4) |
| 3 | Empowerment of personal and context adaptation strategies | 2,378 (6.1) |
| 4 | Problems with study skills and learning strategies | 1,955 (5) |
| 5 | Other answers | 1,505 (3.8) |
| 6 | Orientation and career services | 756 (1.9) |
| 7 | Insufficient/No knowledge of the service | 611 (1.6) |
| 8 | Didactic tutoring | 527 (1.3) |
| 9 | Socio-relational distress | 417 (1.1) |
| 10 | Psychopathological distress | 220 (.6) |
| 11 | Learning disabilities and disabilities | 78 (.2) |
| Total | | 39,277 (100) |

| Expected target students | | |
|---|---|---|
| 1 | Students with study-related emotional distress (anxiety, fears, etc) | 10350 (26.4) |
| 2 | Students experiencing problems with their study skills | 9,211 (23.5) |
| 3 | Students experiencing temporary psychological distress | 7,871 (20) |
| 4 | Students experiencing interpersonal relationship problems (with friends, family, partners) | 2,782 (7.1) |
| 5 | Students with learning disabilities | 2,674 (6.8) |
| 6 | Students with psychopathological problems | 2,463 (6.3) |
| 7 | Students experiencing relationship problems with some teachers and/or technical/administrative staff | 2,460 (6.3) |
| 8 | Students with disabilities | 1,466 (3.7) |
| Total | | 39,277 (100) |

continuous variables that explain most of the heterogeneity, or *inertia* in this context, in the data (analogous to the principal components). As for PCA, the first axis is the most important dimension in terms of the amount of explained inertia, the second axis the second most important, and so on. Specifically MCA exploits a table of relative frequencies (Burt table) to map observations across the computed axes where subjects with similar profiles will be placed close to each other.

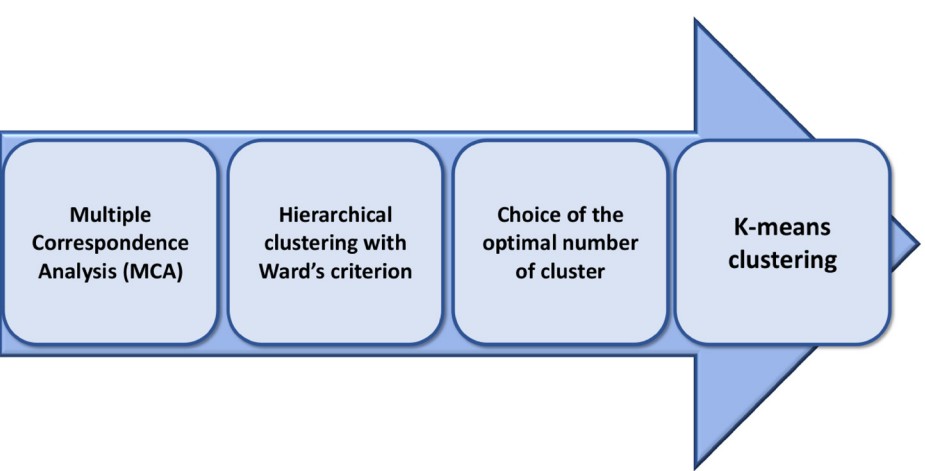

**Fig 1. Flowchart of the proposed methodology.** After a feature selection procedure based on the MCA algorithm, we implemented a hierarchical clustering procedure with Ward's criterion and chose the optimal number of clusters based on the obtained dendrogram. Finally, we used the K-means algorithm to optimize the final partition.

As we wanted to cluster students based on their answers to the third part of the questionnaire, the variables we used to compute the MCA axes, i.e., the *active variables*, were only three: "Expected activities", "Expected target needs", and "Expected target students." For subsequent analysis, we retained as many axes as necessary to explain 90% of the inertia in these data. We used the other five variables, namely "Sex", "Age bin" (we converted age into a categorical variable), "Type of degree", "Student status", and "Disciplinary area" as *supplementary variables*; they are called supplementary because they do not contribute to the definition of MCA axes but nonetheless they characterize the survey participants.

**2.4.2 Hierarchical clustering and inertia gain.**   With MCA we mapped observations(surveyed students) in a continuous space, i.e., we mapped the set of answers a student gave to the third part of the questionnaire to a set of coordinates in the continuous space defined by MCA. In this space, students who gave similar answers are close to each other, and our aim is to build clusters of students who gave similar answers and then to characterize these clusters.

To cluster students we used an agglomerative hierarchical algorithm [38], i.e., a hierarchical method that iteratively merges two elements if their similarity measure is sufficiently high. he specific criterion we used to merge clusters was minimization of inertia, i.e., minimization of the heterogeneity of the resulting clusters (see more details below). The metric that quantifies the distance between pairs of elements is the Euclidean distance ($L_2$ *norm*):

$$d_E(X, Y) = \sum_{j=1}^{N} \sqrt{(x_j - y_j)^2} \tag{1}$$

where $x_j$ and $y_j$ are coordinates of two data points, $X = (x_1, x_2, \ldots, x_N)$ and $Y = (y_1, y_2, \ldots, y_N)$, respectively. The criterion used to measure the similarity between clusters, i.e., the linkage criterion, is based on Ward's inertia minimization principle. Ward's linkage aims at minimizing the total within-cluster inertia $W$, i.e., the heterogeneity within clusters of observations (or, equivalently, at maximizing the between-cluster inertia $B$, given that the total inertia $T = W + B$ is a constant [39]). Ward's method starts from singletons (or clusters that contain only one observation), which implies a state of minimum total within-cluster inertia $W = 0$, and proceeds iteratively. At each step, it merges two clusters, thus increasing the within-cluster inertia

of the resulting partition, as clusters become larger and therefore more heterogeneous. It iteratively merges the pair of clusters that will result in the minimum increase in $W$, compared to any other pair of clusters [40], until all data are clustered into one big cluster with $W = T$ and $B = 0$. These iterations produce a hierarchical tree. Choosing where to cut the tree or equivalently choosing the optimal number of clusters is somewhat arbitrary and is influenced by the specific characteristics of the data. We selected the optimal number of clusters $K$ based both on the decrease in between-cluster inertia $B$ (i.e., on the length of the branches of the dendrogram) and on the interpretability of the obtained clusters.

**2.4.3 K-means clustering.** Once we set the optimal number of clusters using the hierarchical procedure described above, we refined the obtained partition and made it more robust using a few iterations of the K-means method.

The K-means method [41] is a popular clustering procedure that aims at partitioning quantitative observations into $K$ clusters, where $K$ is a given number, so as to minimize within-cluster inertia. In general, the K-means algorithm is efficient, fast, and computationally inexpensive. Given an initial set of $K$ centroids (or means), the algorithm proceeds iteratively by assigning each observation to the nearest centroid, i.e., the centroid with the least squared Euclidean distance, defined in (1). Then it recomputes the centroids of the obtained partition until the algorithm converges. The algorithm can converge to suboptimal centroids as the problem it is trying to solve is NP-hard; an approximate solution is often sought by running the algorithm multiple times with random initial states and retaining the solution with minimal variance. A sensible choice of the initial set of centroids, such as the partition obtained with Ward's method, can substantially increase the probability that the algorithm will converge to the optimal solution.

# 3 Results

## 3.1 Preliminary analyses

We preliminarily reported some descriptive statistics characterizing students with respect to their answers to the second section of the survey with multiple-choice questions. When asked what the target population of UPC is, almost half of the students (46.4%, N = 18,207) correctly identified the target population as university students, but almost the same amount did not seem to have a clear idea as 44.7% (N = 17,563) of the participants chose "None of the previous categories/I don't know." A very low percentage (<4.7%) identified families or academics as the target population of the service.

When asked if they had ever heard about psychological counseling on campus, 71.7% (N = 28,158) of the total sample answered that they had never heard about it, 24% that they had rarely heard about it (N = 9,433), and only 1.4% and 0.7% (N = 531 and 132) that they had frequently and very frequently heard about it, respectively.

Of the total sample, only 11.3% (N = 4,400) reported that they had thought of using the service, while the remaining never thought of doing so (N = 34,837, 88.7%). When asked why, among those who had no intention to use the service, the most frequent answers were not being aware of the service (34.9%) and having no need of it (32.9%). Among those who thought of using the service, the most frequent reasons were a condition of emotional distress (36.1%), followed by lack of strategies to cope with the academic context (10.2%).

Lastly, only 1.4% of the sample (N = 545) had actually used the service, while the remaining had not (N = 38,732, 98.6%). When asked why not, this subsample answered mainly because they were unaware of it (43.4%). Among those accessing the service (N = 545), the main reason for having done so was because they were experiencing emotional distress (27.9%).

### 3.2 Main analyses

**3.2.1 Research question 1. What do students expect from UPC in terms of target population, needs addressed, and activities?**   The first aim was to explore students' expectations of psychological counseling in the academic context, in particular to explore which target population they expected a counseling service to address, which needs they expected it to address, and which intervention strategies it implements.

About one quarter of the respondents identified students with study-related emotional distress (26.4%) and students experiencing problems with their study skills (23.5%) as the target students of an on campus counseling service; about 20% identified students experiencing temporary psychological distress, and the remaining categories received less than 8% of the choices (see Table 2).

When asked about the types of needs targeted by UPC, more than half of the respondents provided no answer or reported to not know which needs were addressed by the service (55.1%); 23.4% of the sample identified personal experience of emotional distress as a target need, and 6.1% a lack of personal coping strategies for the context. Less than 5% of the respondents selected the other categories of need, as listed in Table 2.

As to the expected activities, more than half of the respondents provided no answer or reported to not know which activities were offered by the service (52.7%); 20.2% expected the service to provide a space for sharing and managing personal distress, while the remaining categories listed in Table 2 were selected by less than 4% of the students.

In summary, most students identified students with emotional distress, mostly related to studying, as the typical target of UPC. Also, most students were unaware of both the needs that UPC addresses and the activities that UPC offers. Despite this general result, a minority of students seemed to recognize UPC goals and that is thus represents a valuable resource. A synergistic collaboration can be imagined between this minority of students and appropriate information campaigns and interventions (see Discussion).

**3.2.2 Research question 2. What are the features that characterize students who are willing to use the UPC service (but so far have not) and students who have actually used it.** Secondly, we restricted the focus to two subsamples of the total population of college students: (i) those who have expressed an intention to use it but thus far have not (Group 1) and (ii) those who actually used it (Group 2).

First, we removed students for whom we did not have age or subject-area data (they amounted to less than 1% of the data). As a result, Group 1 consisted of N = 4,038 students and Group 2 of 545 students.

For each group we performed MCA, hierarchical clustering, and K-means partitioning, and then we identified the variables that characterized the resulting clusters. In S1-S4 Tables of the S1 Appendix we report the test results and three descriptive statistics that can help in the interpretation of the results: (i) the percentage of the total population of students who belong to the specific category (column 2), (ii and iii) the percentage of students in the whole group (group 1 or group 2) and the percentage of students in the cluster who belong to the specific category for each cluster.

### 3.3 Group 1

We clustered data using the first 23 MCA components, which account for 89.1% of the inertia (see Fig 2) and Ward's linkage algorithm. The obtained dendrogram is shown in Fig 3. We chose to cut the dendrogram at $K = 6$. Our choice was driven mostly by the interpretability of the resulting clusters as Ward's inertia gain steadily decreases after $K = 2$ (see Figs 2 and 4). A common strategy is to cut the dendrogram where there is a sudden decrease in inertia, i.e., in

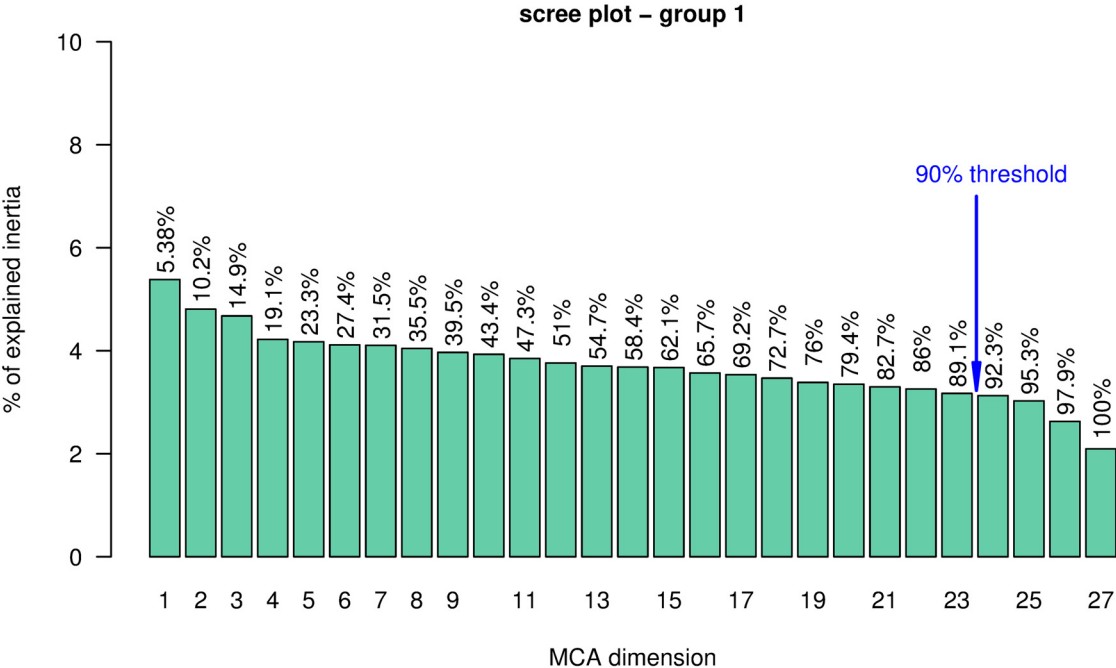

**Fig 2. Scree plot for group 1.** Barplot of the percentage of inertia explained by each of the MCA dimensions. The cumulative percentage of inertia explained is annotated on each bar. The 90% threshold corresponds to 23 MCA components which explain 89.1% of the inertia.

our case at $K = 2$ or $K = 10$. However, $K = 2$ does not generate sensible clusters, as it creates a very small cluster ($C1$) and a very large cluster with all the remaining observations (see Fig 2). Also, $K = 10$ is too big to allow a sensible interpretation of the results. $K = 6$ seems a reasonable choice as after six clusters, the dendrogram consists of repeated subpartitions of cluster $C4$ (in violet in the dendrogram) into a very large set and a very small set.

Four of the six clusters obtained, namely clusters $C1$, $C2$, $C4$, and $C5$ had less than 5% of the data (see Table 3), and we did not consider them for further analyses. The variables that characterize group 1 are "Expected target needs" (p-value < 0.001), "Expected activities" (p-value < 0.001), "Expected target students" (p-value < 0.001), "Disciplinary area" (p-value < 0.05) as obtained with a $\chi^2$ test. The set of variables that characterize the clusters, and are therefore significant in defining the different profiles that emerge from the group, are listed in S1 and S2 Tables of the S1 Appendix.

Cluster 3, labeled "Emotional focus", captures the majority of the students willing to use the service (N = 3,433, 85%): according to them, academic counseling should target students experiencing study-related emotional distress (anxiety, fears, etc.) and students with temporary psychological distress, which was also identified as the main need. Respondents reporting not to know exactly which need the service should target are overrepresented in this cluster. Students in this cluster expect the main intervention strategies to be individual clinical interview sessions, group activities focused on listening, sharing, and coping with personal distress, and they tend to under-rate issues related to disability, learning disorders, psychopathological conditions, and sociorelational distress, as well as activities related to diagnostic assessment, orientation and career service, didactic tutoring, empowering personal and context adaptation strategies, and group interventions. This cluster is transversal to the whole population given that none of the supplementary variables characterize it.

**Dendrogram**

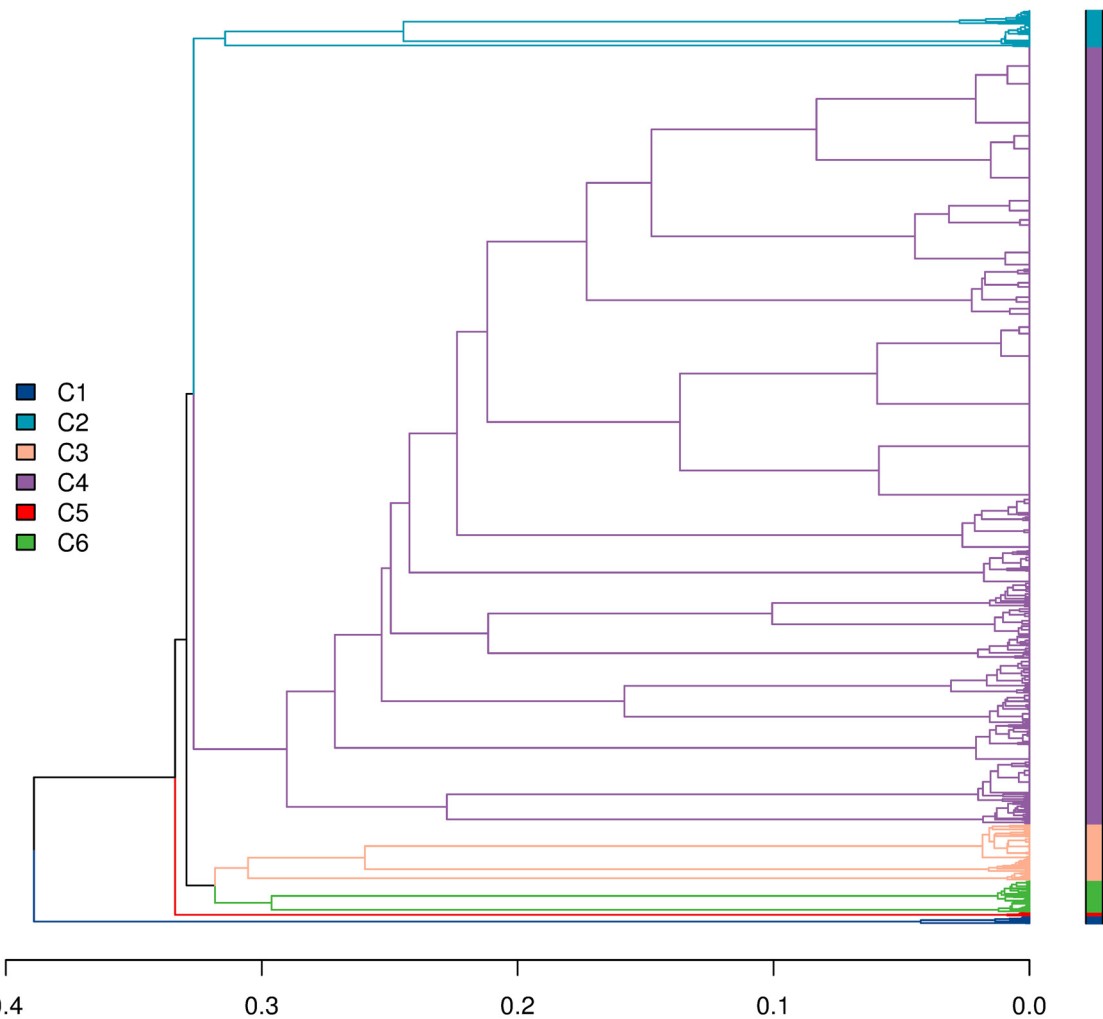

**Fig 3. Dendrogram for group 1.** Different clusters are represented with different colors. We cut the dendrogram at $K = 6$ because for larger values the obtained clusters were sub-partitions of already small clusters or subsequent sub-partitions of cluster $C4$ into pairs of clusters one of which was really small and these additional partitions didn't add to the interpretation of the input data.

Students in cluster 6 (N = 248, 6%), labeled "Psychopathological focus", believe that a counseling service on campus should be directed towards students experiencing relational difficulties with family, friends, and partners. They identify as targets mainly psychopathological conditions and didactic tutoring, while they expect intervention to be mainly at a group level. Disabled students are an underrepresented target population, together with activities such as orientation and career service, clinical interviews, empowerment of personal and context adaptation strategies, and sharing and supporting personal distress. As to the supplementary variables, students from scientific/technological programs are overrepresented in this cluster, while those from sociohumanistic programs are underrepresented.

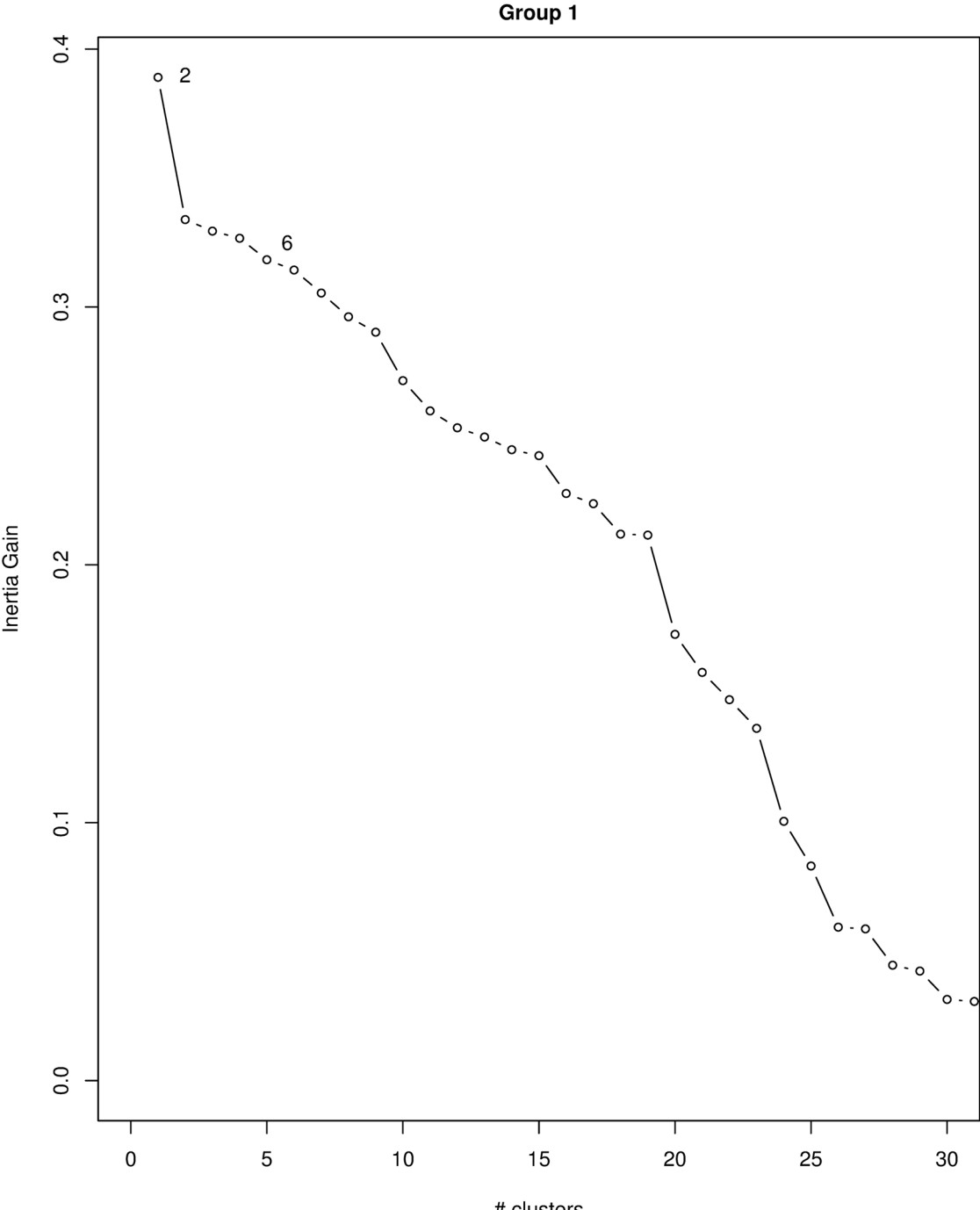

**Fig 4. Plot of the between cluster inertia for group 1.** Between cluster inertia (or inertia gain) smoothly decreases when the number of clusters increases. Sudden decreases of inertia occur at $K = 2$ (too small) and $K = 10$ (too large and difficult to interpret). We chose the intermediate value of $K = 6$ for data interpretability.

### 3.4 Group 2

For group 2, we used the first 22 MCA components, which account for 88.7% of the inertia (see Fig 5) and chose $K = 12$, which left us with three clusters, as many of them have less than 5% of the data and were excluded from the analysis (namely clusters $C1$, $C2$, $C4$–8, $C10$, and

**Table 3. Characterization of the clusters in group 1.**

| C1 | C2 | **C3** | C4 | C5 | **C6** |
|---|---|---|---|---|---|
| 35 | 160 | 3433 | 145 | 17 | 248 |

Number of students belonging to each cluster. We only considered clusters with more than 5% of the data for further analysis (in bold).

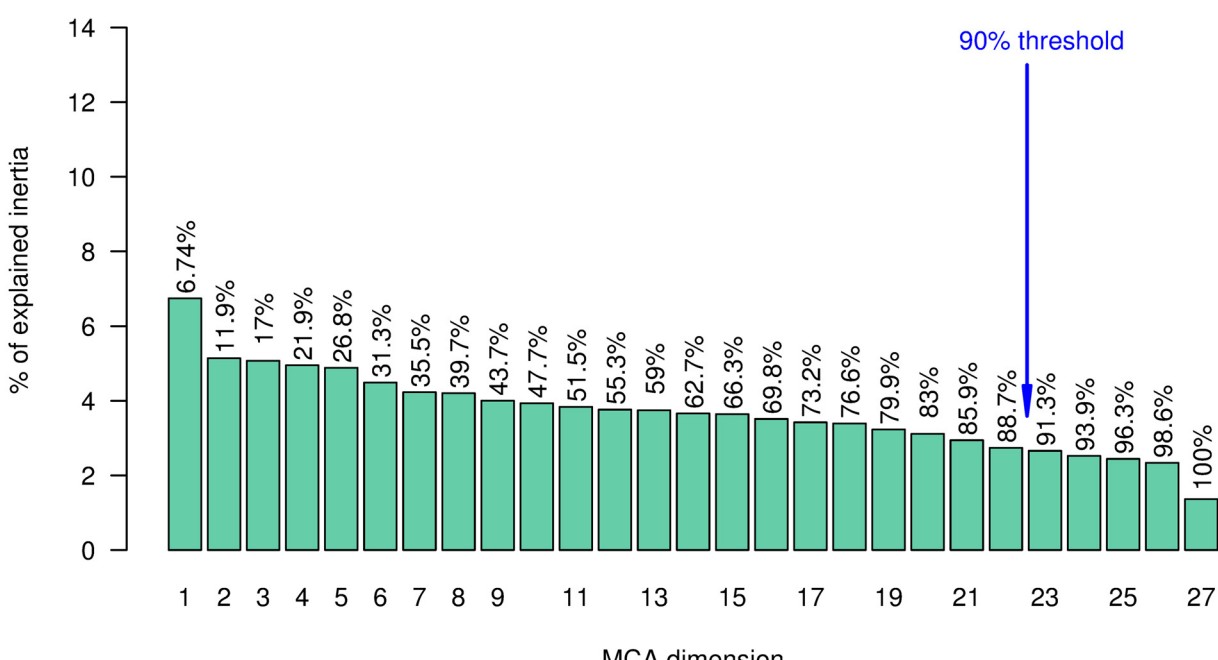

**Fig 5. Scree plot for group 2.** Barplot of the percentage of inertia explained by each of the MCA dimensions. The cumulative percentage of inertia explained is annotated on each bar. The 90% threshold corresponds to 22 MCA components which explain 88.7% of the inertia.

*C*11; see Table 4). As for group 1, we observed a steady decrease in the between inertia after $K = 2$ (see Figs 6 and 7) and again, after $K = 12$, we observed the same scenario as with group 1: mostly, cluster *C*6 gets split into subsequent pairs of small and big clusters. $K = 12$ seems therefore a sensible choice in this case. The variables that characterize group 2 are "Expected target needs", "Expected activities", "Expected target students", and "Student status" with p-value < 0.001 and "Age bin" with p-value < 0.01, as obtained with a $\chi^2$ test. The set of variables that characterize each cluster are listed in S3 and S4 Tables of the S1 Appendix.

**Table 4. Characterization of the clusters in group 2.**

| C1 | C2 | **C3** | C4 | C5 | C6 | C7 | C8 | **C9** | C10 | C11 | **C12** |
|---|---|---|---|---|---|---|---|---|---|---|---|
| 1 | 13 | 247 | 1 | 1 | 1 | 6 | 2 | 40 | 25 | 6 | 202 |

Number of students belonging to each cluster. We only considered clusters with more than 5% of the data for further analysis (in bold).

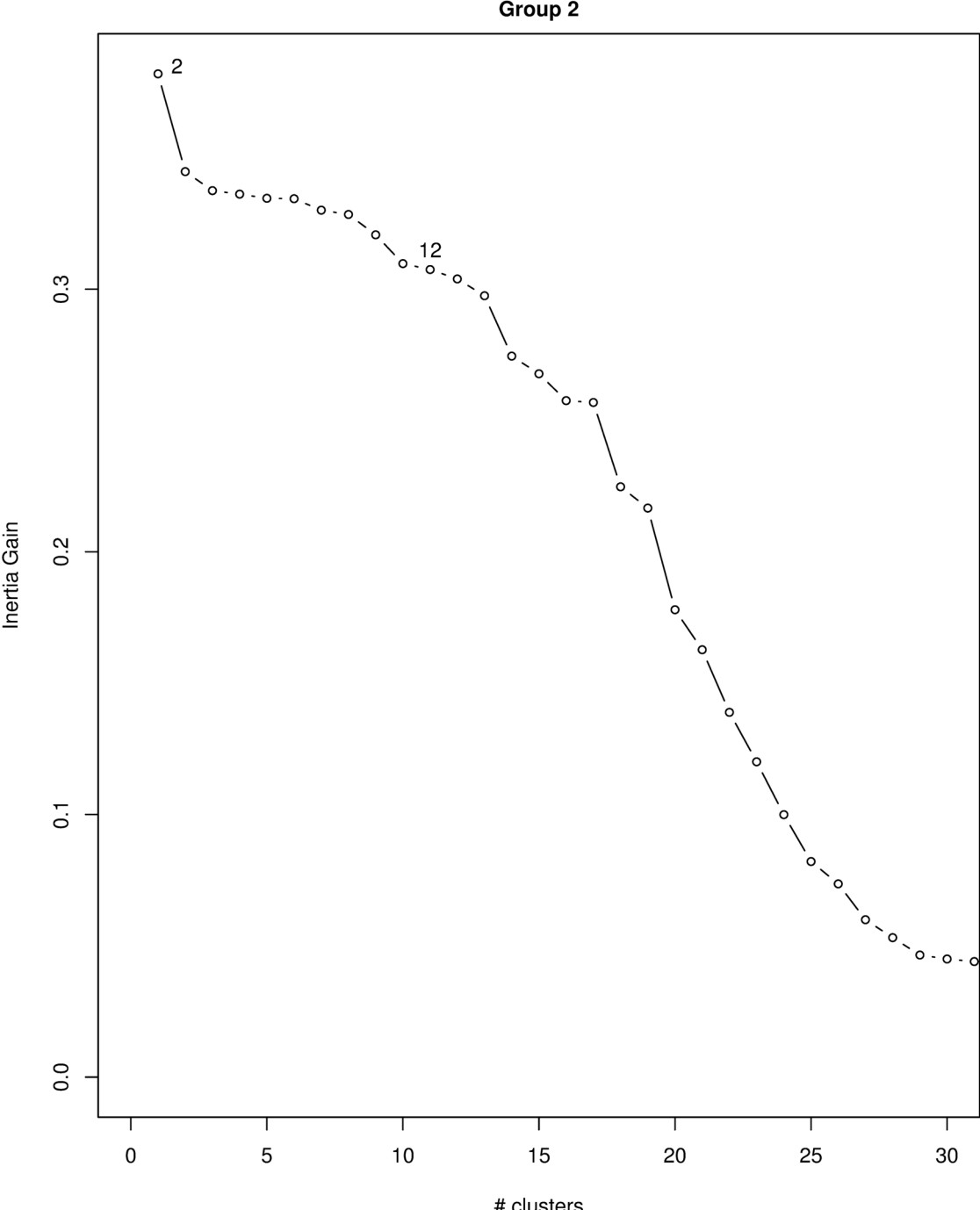

**Fig 6. Plot of the between cluster inertia for group 2.** Between cluster inertia (or inertia gain) smoothly decreases when the number of clusters increases and we cannot identify a clear cutoff. We chose $K = 12$ as a trade-off between inertia gain and interpretability of the obtained clusters.

What mainly characterizes students in cluster 3 (N = 247, 45%), labeled "Clinical focus", is the identification of the population to whom counseling is directed: students with learning disabilities, psychopathological disturbances, students experiencing relationship problems with some teachers and/or technical/administrative staff, and students experiencing problems with

**Dendrogram**

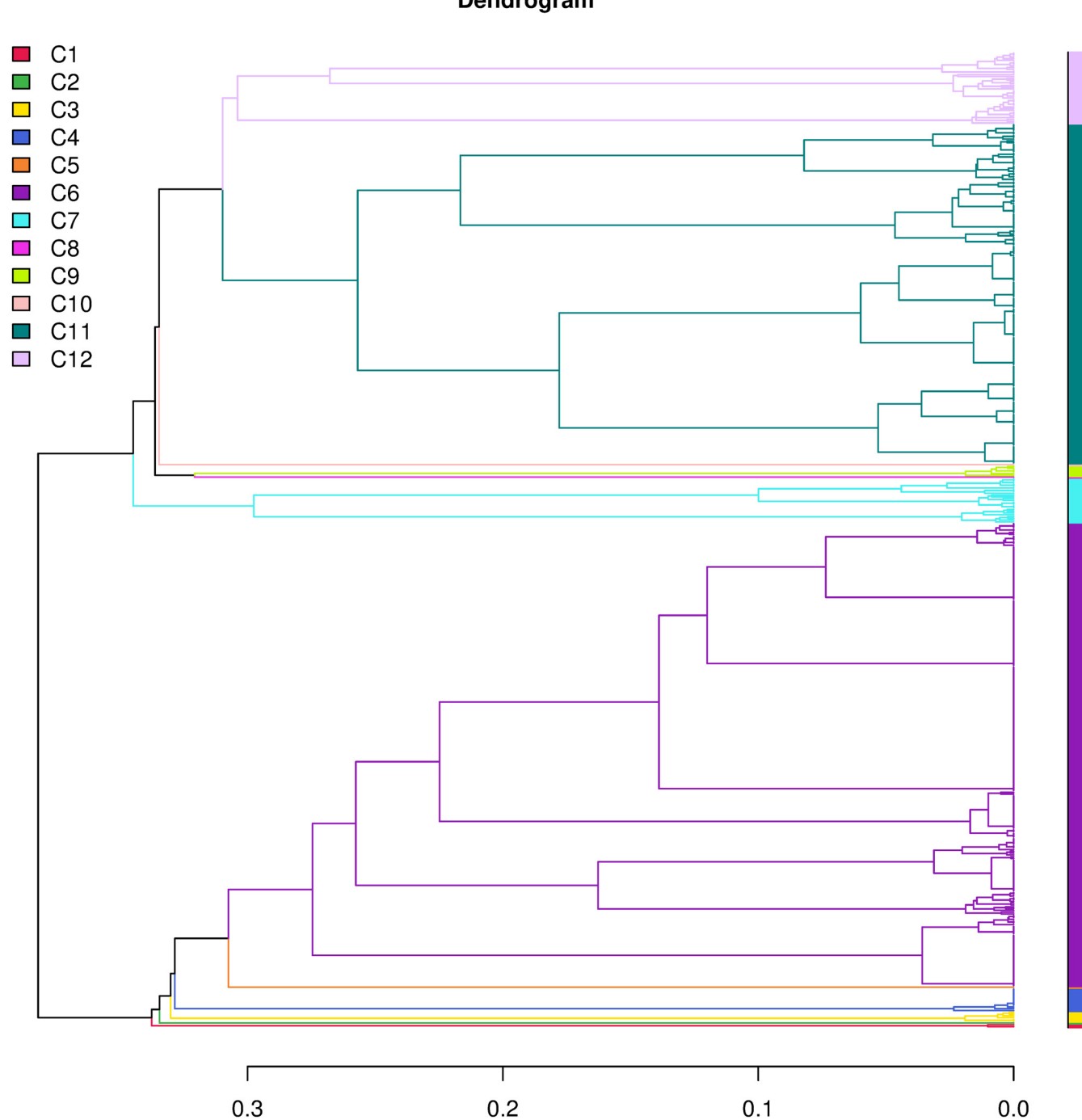

**Fig 7. Dendrogram for group 2.** Different clusters are represented with different colors.

their study skills. Students in this cluster do not seem to have a clear idea of which needs and intervention strategies should characterize a university counseling service, as almost all of the categories for each dimension appear to be underrepresented. As to supplementary variables, individuals in this cluster have an overrepresentation of men, young students, enrolled in a

bachelors' degree, on track with their academic career, and registered in a degree within the health disciplinary area.

Cluster 9 (N = 40, 7%) has an overrepresentation of students who are inactive and have a "Learning focus." It groups students that identify problems with study skills and learning strategies as the need that a counseling service should address through group interventions aimed at strengthening learning strategies and study methods.

The last cluster, number 12 (N = 202, 37%), labeled "Socioemotional focus", identifies three target populations all sharing distress from different sources: from study and academic performance, from interpersonal relationship issues (with friends, family, partners), or from personal psychological distress. As to the needs a counseling service should target, they identify emotional distress and a lack of coping strategies. As to the intervention strategies, they identify individual clinical interviews and sharing and managing personal distress. This cluster has an overrepresentation of students in the second half of their twenties, mainly women, off-track with their academic degree, and attending a single-level degree of five to six years.

In summary, we used clustering analysis and the obtained dendrograms reported in the Supplementary Materials to identify further subgroups of Group 1 and Group 2. Then, after we identified the subgroups, we proceeded with a statistical analysis to characterize them. We observed that students in Group 1 and students in Group 2 had two different mental representations of the service. On one side, students who expressed their intention to use the counseling service but had not yet done so (Group 1), had a less differentiated, narrower and more common perception of the service, mainly adjusted to an emotional and a psychopathological focus, without a clear idea of the needs the service should target. On the other side, students who had accessed the service in the past (Group 2), had a more articulated representation of the service, and expected a service that adapts with flexibility to a variety of target populations and needs including clinical, learning, and socioemotional areas.

## 4 Discussion

The purpose of this study was to try to "ecologically" explore the perspectives on UPC services among a large sample of university students, almost equal to the total population of a large university in Southern Italy. After a brief description of some preliminary information, we addressed this objective by examining two questions: (a) In the general student population, what do students expect from UPC in terms of target population, needs addressed, and activities? (b) Restricting the focus to those students who expressed a positive intention to use the UPC service and to those who already had done so, can we characterize their perceptions and expectations and eventually how do they differ?

In general, the results suggest that only a little more than a quarter of the sample was at least minimally aware of the UPC service offered by the University, while only a few had really significant knowledge of it (about 2%). A minimal percentage of students, less than 2%, had actually accessed the service. Furthermore, among those who had thought about using the service (about one out of ten), they had done so mainly for issues related to emotional distress and a lack of strategies to succeed on their academic path. Overall, these descriptive statistics provide direct observational evidence of a larger trend of underestimation and lack of information of the value of UPC in Italian Universities. As already mentioned in the introduction, the international and Italian literature suggests that approximately 30% of university students report symptoms associated with mental disorders [13–15]; consequently, the fact that only 1.4% of the studied sample had used psychological counseling constitutes a big issue with relevant educational and social implications. The use of counseling services has been, in fact, associated with an increase in academic achievement in both the general and at-risk student population

[42, 43]. The underuse of these services therefore exposes a condition of greater vulnerability due to the educational failure of students, with a consequent risk of increased dropout rates in the academic community. However, about half of the respondents, although largely unaware of the service and without expectations regarding its use, seemed to adequately recognize university students as the recipients of the service, opening a positive scenario for a form of awareness aimed above all at explaining the advantages and opportunities offered by UPC and not only at identifying and defining users.

Accordingly, it must be underlined that this investigation took place at, and influenced, the beginning of a massive reorganization of the psychological service in the university context where the research took place. These findings were very useful at the time to guide the implementation of a massive communication campaign through the University's website about the availability of the service, which resulted in an increase in monthly requests by 530% in the first three months after the new service was implemented and empowered. Besides the local impact of these results, at a more general level, these results show the importance of keeping students informed and updated about the availability of the service in order to strengthen their awareness about their own needs and in order to encourage them to ask for support. The more invisible and uncommon such a service might be perceived to be by young adults, the less they will confront with their needs and find a way to ask for support.

When moving to the first research question, a picture emerged of what students perceived as the main targets of a counseling service: students characterized by study-related emotional distress (with symptoms of anxiety, fear, etc.), problems with study skills, and/or temporary psychological distress. Consistently, the needs targeted by the psychological counseling service are expected to be mainly related to overcoming personal emotional distress, strengthening personal adaptation strategies in the university context, and supporting study skills and learning strategies. Finally, in terms of expectations regarding the activities implemented, the service is expected to be active, especially in providing a safe context in which to share personal distress and find support to manage it and a context for individual clinical interviews and educational tutoring. This map of expectations reinforces the idea that at least a good part of the students are able to recognize the specific mission of UPC. These students represent a resource for the promotion of mental health in the university context. According to Biasi [4], in fact, a counseling process is undertaken when a person who has recognized that they need help to get out of a problematic situation turns to a specific service to manage the problem themself more effectively. Counseling would therefore be aimed at encouraging the recognition of students' resources, strengths, and abilities in order to help them solve psychological or emotional problems. It is generally structured in the form of interviews in which the student is also invited to reflect on the choice of educational path, on their ambitions and aspirations, or on the methods of study and the difficulties encountered in interacting with the university, according to the "question" with which the student addresses the service.

Furthermore, the main issues that are the target of the counseling intervention generally refer to difficulties in studying, difficulties in taking exams, difficulties in completing one's own path, doubts about choosing an academic path, difficulties in entering the university context, and personal problems influencing the study and concerning issues related to family or personal health problems, which can drastically limit the possibility of participation in academic life and the willingness to study. This perspective, as already suggested, seems to correspond well to the set of expectations expressed by actively responding students (about half of the total sample). This suggests that it could be very advantageous for university counseling centers to collaborate with these students on awareness and prevention campaigns to increase their impact given that the students themselves could represent concrete examples of use, albeit cooperative, of the university counseling services [7], countering the possible barriers of

embarrassment and shame that are often linked to the stigma surrounding using mental health services [19]. In addition, the collaboration would offer the opportunity for university consulting services to constantly acquire a wide range of information on the mental health of students over time and to enhance the development of new ideas among them to change the culture of their university context. Consequently, the collaboration would become an intervention itself [7], both in terms of the personal health of the students engaged in common actions and in terms of increased public awareness about the importance of students' mental health.

However, it must be also emphasized that when looking at the expected target needs and activities among the whole sample, it emerged that students expect from UPC more than they are supposed to offer, which seems to suggest a quite undifferentiated perception of the different services available on campus: for instance, difficulties in choosing an academic path should be addressed by the orientation service, while difficulties with complex subjects should be addressed by the tutoring service. All Italian universities are expected to have these two services, according to the Italian Ministry of Education, which distributes specific funding to support them. The undifferentiated expectation that emerged from this survey highlights again the importance of a massive and clear communication campaign to help students distinguish and identify the different services available on campus. These findings also suggest that besides dealing with psychological distress, psychological interventions could also aim to strengthen study methods, help students manage anxiety related to exams and deadlines, and organize their time in a more strategic way. All of these interventions were, in fact, implemented during the above-mentioned reorganization of the psychological service in our university context following these results.

With reference to the second research question, among students who expressed a positive intention to use the psychological counseling service but had not done so at the time of the survey, two main student profiles emerged, which together represent nine out of ten participants in the subsample. In particular, the "emotional focus" profile was predominant and characterized by the idea that the university counseling service is aimed above all at students who experience emotional distress related to study and general psychological distress and that the needs that must be met are also linked to overcoming these distressing conditions. In addition, the interventions that the subjects in this profile expected were individual clinical interviews and group activities based on listening and sharing. The second profile, defined as "psychopathological focus", was made up of students who believed that the counseling service is essentially aimed at students who are experiencing relational difficulties with family, friends, and partners and that the needs to be met mainly concerned the overcoming of psychopathological conditions. In addition, they expected a type of intervention that falls into the category of the didactic tutor, that is, the coaching of a colleague on the university path. Both the two groups seem to refer to a fairly widespread and common representation of UPC services, which are seen as centers to ask for help in the case of psychological or emotional problems. In the first case, however, there seemed to be a tendency to refer to more individual problems, while in the second case, to relational difficulties with more relevant implications in terms of mental illness. The fact that this second profile prevailed among students of more scientific-technological courses than sociohumanistic ones made us reflect on the fact that the type of course can influence the perception that students have of psychological counseling, based on their academic experience. In fact, science-technology students usually attend courses with a low number of students, in which it is probably more difficult to establish a significant number of relationships with colleagues and partners. This could, consequently, turn into a greater need to be referred to aid services that specifically address such difficulties. Besides this feature, no other supplementary variable was particularly distinguished across the clusters, which seems to

suggest that perceptions and expectations are quite transversal across courses, ages, sexes, and study fields.

In the group of students who had used the psychological counseling service, three main profiles emerged, which grouped together nine out of ten subsample participants. The first profile was named "clinical focus" and was characterized by students with learning disabilities, psychopathological disorders, relationship problems with teachers and university staff, and problems with study skills. This group was not characterized by specific expectations about any need or activity of a university counseling service, while it was strongly characterized by involving men, the young age, and the frequency of courses in the health area. The second profile called "socioemotional focus" identified a population characterized by problems of distress, linked to academic performance, relational problems, and individual psychological situations. As regards the needs that, according to this group of subjects, a counseling service should satisfy, the overcoming of both emotional distress and the lack of strategies to adequately cope with university courses were identified. The expected interventions concerned individual clinical interviews and activities for sharing and managing personal distress. Furthermore, this profile was characterized by a greater number of women, off-track with their academic degree, and attending a four- or five-year degree course. Finally, the third profile, numerically less relevant, was defined as "learning focus" and was mainly made up of students who identified problems with study skills and learning strategies as the needs that a counseling service should satisfy through activities in line with these needs, for example by making the study method more effective. From a general view of the three profiles, it emerges that the students who have used the university counseling service have a more articulated and flexible vision of the service itself; it represents an opportunity to face more clearly clinical problems, to seek emotional-psychological support, or to have support from a learning point of view. When one enters directly into contact with the university counseling service, therefore, this seems to adequately guide the students' perception towards the different areas subject to counseling, guaranteeing recipients adequate support for the incoming "demand" and, in turn, reducing study delays in general, which represent real risk factors for university dropout.

Combined together, the results of the two subgroups of students (those who are positive but have not used university counseling and those who have used it) seem to suggest that in the first group there was a more "superficial" and "common" representation of the UPC, while in the second there was a more faithful view of what it represents. This fact underlines the need for a greater student awareness effort. As already mentioned, one possibility is the involvement of active students in the services provided through psychological counseling; however, the use of IT channels, in view of their ever-growing popularity among students, should not be underestimated. The percentage of responses obtained through our survey (about 90% of the entire university reference population) demonstrates how the use of the university's intranet channels, but also of more general channels such as YouTube, Instagram, or simple e-mail, can effectively broaden the impact of seminars, workshops, and brochures that have usually been the preferred dissemination channels. Furthermore, on the basis of the profiles obtained, the need for a variety of campaigns, both to raise awareness and to intervene and prevent some specific characteristics of the student population, should not be underestimated. For example, when addressing students of scientific-technological courses, it would seem more effective to communicate and intervene by favoring the sphere of social relations, while with women, off-track with their academic degree, and attending a single-cycle degree course, it would seem more appropriate an approach focused on personal psychoemotional issues. A characterization of counseling services in this sense would have the advantage of reaching more specific groups of students, guaranteeing them a greater perception of closeness, and therefore generating greater motivation to use the service. It is through reflections of this kind that

psychological counseling services can effectively fulfill that strategic function aimed at contributing to a reduction in both university dropout levels and access gap, as previously discussed in the introduction.

In any case, the results of this study should be interpreted in the context of certain limitations. The data were collected from a single university and in a single regional context. The research, therefore, captures well the specificities and peculiarities of one and the other, but this limits the generalizability of the interpretations made. Our survey was also based on a non-standardized questionnaire, built ad hoc to answer the research questions presented here. This makes it difficult to compare with other studies that have used different tools and measures. Furthermore, the use of open questions, coded through a bottom-up coding system, poses problems with respect to the contribution of this work in the context of theoretical advancement, although this paper did not have this intent. Despite this, the observational and highly ecological nature of our work has a good chance of stimulating further reflections and research questions, which should increasingly be examined through mixed-method approaches which have the advantage of firmly integrating qualitative aspects with quantitative analyses.

Notwithstanding these limitations, the study makes a new contribution by exploring UPC through a bottom-up approach, that is through students' perceptions and expectations of the service and testing whether this approach matches the top-down one defined by scientific and professional associations' guidelines. Our results seem to provide a partial confirmation, as the top-down professional and scientific guidelines on psychological counseling in academic contexts overlap with students' perceptions and expectations in a way largely independent of their age, sex, academic courses, and profile. On the other hand, it is also true that regarding many themes, about half of the respondents did not have a clear idea or revealed a common and undifferentiated idea about UPC, which could in part explain the existing gap between young adults' need for mental health counseling and the effective use of the counseling service.

## 5 Conclusions

In recent years, UPC has received much attention as a possible means of promoting mental health, academic success, and, in the long term and on a large scale, the growth of society. Our study has tried to contribute to the existing literature on the implementation and dissemination of the practice of psychological counseling in the university context, with a bottom-up approach complementary to the usual top-down one that follows the guidelines of scientific-professional associations for counseling. The results reveal that a large majority of students do not have a clear idea both about the needs that UPC should address and the activities it should offer. However, about half of them were able to adequately identify university students as the target population of UPC. These latter students can be considered an important resource for active involvement in awareness and prevention campaigns aimed at their peers in order to increase the use of UPC and consequently reduce the impact of mental health problems among university students. Considering two specific subsamples of the study participants, namely (i) those who only expressed their intention to use the counseling service and (ii) those who had already used it, different student profiles are found in the two groups. For the first group, a profile with an "emotional" focus and one with a "psychopathological" focus were identified; in the second group, a profile with a "clinical" focus, one with a "socioemotional" focus and one with a "learning" focus were identified. This picture of the results suggests that in the former there is a fairly "superficial" and "common" representation of UPC, while in the latter this representation is more "articulated" and "flexible." This second series of results confirms the need for intense communication campaigns, even among students who intend to use counseling but have not yet done so. Such campaigns should better clarify the specific

characteristics of the service and could use IT channels, which are widely used among students. Overall, the findings of this study suggest the importance of adopting student-centered strategies both to reach a wider audience and to identify, and therefore respond, to students' specific expectations and needs. Furthermore, these findings align with top-down approaches to identify the variables that are important for a successful UPC service. Nonetheless, further studies adopting both approaches are needed and expected in this research area.

## Supporting information

**S1 Appendix. Additional tables.** Tables S1–S4 report test results and three descriptive statistics that aid in the interpretation of the results.
(PDF)

## Author Contributions

**Conceptualization:** Pasquale Musso, Gabrielle Coppola.

**Formal analysis:** Ester Pantaleo.

**Methodology:** Ester Pantaleo, Domenico Diacono.

**Software:** Ester Pantaleo, Domenico Diacono.

**Visualization:** Ester Pantaleo, Alfonso Monaco.

**Writing – original draft:** Pasquale Musso, Gabrielle Coppola, Ester Pantaleo, Alfonso Monaco.

**Writing – review & editing:** Pasquale Musso, Gabrielle Coppola, Ester Pantaleo, Nicola Amoroso, Caterina Balenzano, Roberto Bellotti, Rosalinda Cassibba, Domenico Diacono, Alfonso Monaco.

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
