## [Decision Letter · Decision Letter 0]

29 Nov 2021

PONE-D-21-25129Psychological counseling in the Italian academic context: expected needs, activities and target population in a large sample of studentsPLOS ONE

Dear Dr. Pantaleo,

Thank you for submitting your manuscript to PLOS ONE. After careful consideration, we feel that it has merit but does not fully meet PLOS ONE’s publication criteria as it currently stands. Therefore, we invite you to submit a revised version of the manuscript that addresses the points raised during the review process. The reviewers raised concerns regarding the clarity of methods section, and the data analysis (see attached file). Both reviewers have questioned the contributions of the paper. Therefore, I invite you to respond to such concerns and make the necessary changes to the paper.

We look forward to receiving your revised manuscript.

Kind regards,

Mohammed Saqr, Ph.D

Academic Editor

PLOS ONE

Journal Requirements:

2. You indicated that ethical approval was not necessary for your study. Could you please provide further details on why your study is exempt from the need for approval and confirmation from your institutional review board or research ethics committee (e.g., in the form of a letter or email correspondence) that ethics review was not necessary for this study? Please include a copy of the correspondence as an ""Other"" file.

3. Please change "female” or "male" to "woman” or "man" as appropriate, when used as a noun (see for instance https://apastyle.apa.org/style-grammar-guidelines/bias-free-language/gender).

Reviewers' comments:

Reviewer's Responses to Questions

**Comments to the Author**

1. Is the manuscript technically sound, and do the data support the conclusions?

Reviewer #1: Yes

Reviewer #2: Yes

2. Has the statistical analysis been performed appropriately and rigorously? 

Reviewer #1: I Don't Know

Reviewer #2: Yes

3. Have the authors made all data underlying the findings in their manuscript fully available?

Reviewer #1: No

Reviewer #2: Yes

4. Is the manuscript presented in an intelligible fashion and written in standard English?

Reviewer #1: No

Reviewer #2: Yes

5. Review Comments to the Author

Reviewer #1: The authors report the results from the survey conducted in a college in Italy in relation to the use of the university psychological counseling. I was initially excited to learn about ways to improve mental health on college campus, but the results reported in the study are unsurprising and appear to be not so helpful in solving actual problems involving mental health issues in colleges. To summarize the findings, I'll just quote from the paper; "a picture emerged of what students

perceived as main targets of a counseling service: students characterized by study-related emotional distress (with symptoms of anxiety, fear, etc.), by problems with study skills, and/or by temporary psychological distress." None of these is surprising. And the implications from this finding, "Consistently, the needs targeted by the psychological counseling service are expected to be mainly related to overcoming personal emotional distress, strengthening personal adaptation strategies and the university context and supporting study skills and learning strategies." are also unsurprising. The other finding from the study is, "the service is expected to be active especially in providing a safe context where to share personal distress and find support to manage it, individual clinical interviews and educational tutoring." Again, that's something to be expected. It's unclear how useful these findings would be for other colleges in improving their own psychological counseling.

Reviewer #2: A few concerns need to be addressed-Check the uploaded report for details. More emphasis on making the data analysis as clear as possible so that if the study is replicated elsewhere its reliability will be high.

6. PLOS authors have the option to publish the peer review history of their article (what does this mean?). If published, this will include your full peer review and any attached files.

Reviewer #1: No

Reviewer #2: No

---

## [Author Response · Author response to Decision Letter 0]

15 Feb 2022

We attached a file titled "Response to Reviewers" which contains detailed answers to all points raised by the reviewers.

---

## [Decision Letter · Decision Letter 1]

30 Mar 2022

Psychological counseling in the Italian academic context: expected needs, activities and target population in a large sample of students

PONE-D-21-25129R1

Dear Dr. Pantaleo,

We’re pleased to inform you that your manuscript has been judged scientifically suitable for publication and will be formally accepted for publication once it meets all outstanding technical requirements.

Kind regards,

Mohammed Saqr, Ph.D

Academic Editor

PLOS ONE

Additional Editor Comments (optional):

Reviewers' comments:

Reviewer's Responses to Questions

**Comments to the Author**

1. If the authors have adequately addressed your comments raised in a previous round of review and you feel that this manuscript is now acceptable for publication, you may indicate that here to bypass the “Comments to the Author” section, enter your conflict of interest statement in the “Confidential to Editor” section, and submit your "Accept" recommendation.

Reviewer #2: All comments have been addressed

2. Is the manuscript technically sound, and do the data support the conclusions?

Reviewer #2: Yes

3. Has the statistical analysis been performed appropriately and rigorously? 

Reviewer #2: Yes

4. Have the authors made all data underlying the findings in their manuscript fully available?

Reviewer #2: Yes

5. Is the manuscript presented in an intelligible fashion and written in standard English?

Reviewer #2: Yes

6. Review Comments to the Author

Reviewer #2: The authors addressed and responded to literally all comments and questions previously raised to my satisfaction. As the manuscript stands at the moment, it has wider readability.

7. PLOS authors have the option to publish the peer review history of their article (what does this mean?). If published, this will include your full peer review and any attached files.

Reviewer #2: **Yes: **SCHOLASTIC ADELI

---

## [Editor Report · Acceptance letter]

1 Apr 2022

PONE-D-21-25129R1 

Psychological counseling in the Italian academic context: expected needs, activities, and target population in a large sample of students 

Dear Dr. Pantaleo:

I'm pleased to inform you that your manuscript has been deemed suitable for publication in PLOS ONE. Congratulations! Your manuscript is now with our production department. 

Kind regards, 

on behalf of

Dr. Mohammed Saqr 

Academic Editor

PLOS ONE